# Updates and Challenges in ENS Cell Therapy for the Treatment of Neurointestinal Diseases

**DOI:** 10.3390/biom14020229

**Published:** 2024-02-16

**Authors:** Takahiro Ohkura, Alan J. Burns, Ryo Hotta

**Affiliations:** 1Department of Pediatric Surgery, Massachusetts General Hospital, Harvard Medical School, Boston, MA 02114, USA; tohkura@mgh.harvard.edu (T.O.); alan.burns@ucl.ac.uk (A.J.B.); 2Stem Cells and Regenerative Medicine, Great Ormond Street Institute of Child Health, University College London, London WC1N 1EH, UK

**Keywords:** cell therapy, neurointestinal diseases, cell transplantation, enteric neural crest-derived cells, regenerative medicine

## Abstract

Neurointestinal diseases represent a significant challenge in clinical management with current palliative approaches failing to overcome disease and treatment-related morbidity. The recent progress with cell therapy to restore missing or defective components of the gut neuromusculature offers new hope for potential cures. This review discusses the progress that has been made in the sourcing of putative stem cells and the studies into their biology and therapeutic potential. We also explore some of the practical challenges that must be overcome before cell-based therapies can be applied in the clinical setting. Although a number of obstacles remain, the rapid advances made in the enteric neural stem cell field suggest that such therapies are on the near horizon.

## 1. Introduction

The enteric nervous system (ENS) is the extensive network of neurons and glia within the wall of gastrointestinal (GI) tract that regulates all aspects of GI function, including motility, sensation, absorption, secretion, and immunity [1,2]. Abnormalities of the ENS, which can result from developmental defects, inflammation, infection, or age-associated neurodegeneration, lead to serious and life-threatening GI dysfunctional conditions [3,4,5,6]. These so-called neurointestinal diseases (NIDs) include Hirschsprung disease (HSCR), esophageal achalasia, Chagas disease, gastroparesis, chronic intestinal pseudo-obstruction, slow transit constipation, and more [1]. They cause serious morbidity and reduced quality of life and result in significant healthcare costs. More than 40% of persons worldwide are reported as having functional gastrointestinal disorders [7], and in the US in 2018, gastrointestinal health care expenditures totaled USD 119.6 billion [8]. The direct annual cost of ambulatory clinic visits alone for chronic symptomatic functional bowel disorders was reported as USD 358 million [9]. While functional GI disorders have many causes, abnormalities in the ENS are among the most important. Despite the prevalence and severity of NIDs, current therapeutic options are inadequate and limited. They aim to palliate the associated symptoms, such as nausea, vomiting, constipation, and abdominal pain, without addressing the underlying pathophysiology by directly targeting the enteric neuronal abnormalities. Cell therapy offers a novel approach for patients with NIDs by repairing or replacing the defective ENS, re-establishing normal neural innervation, GI function, and the restoration of gut homeostasis [10,11].

Cell therapy refers to the transfer of autologous or allogeneic cellular material into a patient for medical purposes [12]. The concept in cell therapy is that replacing damaged or dysfunctional cells with healthy cells will restore lost function and therefore ameliorate associated symptoms [13]. The first practices of cell therapy date back to the late 1880s when animal tissue extracts were injected into patients in an attempt to suppress the effects of aging [14]. Today, cell therapy continues to evolve with numerous investigations for clinical safety and efficacy underway and with a global market size estimated to expand from USD 9.5 billion in 2021 to USD 23.0 billion in 2028 [15]. Cell therapy includes stem-cell- and non-stem-cell-based, unicellular, and multicellular therapies, with different immunophenotypic profiles, isolation techniques, mechanisms of action, and regulatory levels [16]. It spans multiple therapeutic areas, such as regenerative medicine, immunotherapy, and cancer therapy. Here, we will focus on the progress made over the last few decades towards the development of cell replacement therapy as a novel, curative treatment for NIDs, exploring topics including different cell sources, modifications that could be made to the cells, or the local environment into which they are transplanted to help improve ENS restoration/repair, and how to address ongoing challenges facing the successful development of an efficacious ENS cell therapy.

## 2. Overview and Updates in ENS Cell Therapy Research

### 2.1. Cell Sources

NIDs are primarily caused by abnormalities or dysfunction of the ENS. The ENS originates from the neural crest. During development enteric neural crest-derived cells (ENCDCs) colonize the entire length of the GI tract and differentiate into neurons and glial cells [17,18]. A small population of ENCDCs resides within the gut wall even after the ENS is fully developed and remains undifferentiated, serving to replace damaged or lost ENS [19,20,21]. Based on seminal studies in 1999 that identified and characterized these cells and showed that they are multipotent and capable of colonizing aganglionic gut in organ culture, Vassilis Pachnis and colleagues first raised the idea that “The ability of a phenotypically defined population of enteric neural crest cells to colonize the mesenchyme of aganglionic gut and differentiate into mature neurons and glia suggests the feasibility of alternative therapeutic approaches for the treatment of severe cases of Hirschsprung’s disease which are based on transplantation of ENS progenitors” [22]. The success in these early studies showing the engraftment, migration, and neuroglial differentiation of ENCDCs following transplantation to embryonic aneural hindgut [22,23] paved the way for studies assessing the potential of cell therapy for the treatment of NIDs. ENCDCs or enteric neural stem cells (ENSCs) have subsequently been isolated from laboratory animals [21,24,25] and humans [26,27,28], and a number of studies have demonstrated the successful establishment of functioning enteric neurons following an allogenic transplant to the intestine of postnatal mice in vivo [29,30], which led to a functional improvement of GI motility in animal model of NIDs [31,32]. Due to this extensive work with ENCDCs, these cells are considered a promising autologous cell source for cell replacement therapy in NID patients [33] (Table 1). However, in addition to ENCDCs, increasing attention has more recently been drawn to other novel cell types that could be used for NID therapy, namely induced pluripotent stem cells (iPSCs) and Schwann cells (SCCs).

Pluripotent stem cells (PSCs) are somatic cells reprogrammed to a pluripotent state by the forced expression of several transcriptional factors, the so-called Yamanaka factors [34]. These cells are regarded to possess significant advantages for clinical application, including the potential of deriving cells from patients themselves (autologous PSCs), or ultimately having a universal (allogeneic) cell approach, the high accessibility of tissue from which to generate cells (e.g., from skin biopsy), the ability to differentiate cells into multiple cell types/lineages, and unlimited self-renewal capacity [34]. Studer and his colleagues established and optimized protocols for the induction of a neural crest lineage from PSC [35,36,37] and successfully derived enteric neural crest (ENC) precursors from PSCs [38,39]. Interestingly, when these PSC-ENC precursors were transplanted into the Ednrb−/− mouse model of HSCR they extensively colonized the colon, demonstrated neuroglial differentiation, restored peristaltic movement in the colon, and increased animal survival [38,39] (Table 1).

**Table 1 biomolecules-14-00229-t001:** Potential cell sources for the use of cell therapy for neurointestinal diseases.

Target Diseases	Cell Sources	References
Intestinal aganglionosis (Hirschsprung disease)	Human gut-derived neural progenitors	Pan et al., 2022 [40], Cheng et al., 2017 [41], McCann et al., 2017 [32], Rollo et al., 2015 [28], Metzger et al., 2009 [42], Lindley et al., 2008 [26], Hetz et al., 2008 [26].
	Human neural progenitors from non-gut tissues	Yoshimaru et al., 2022 [43], Thomas et al., 2020 [44].
	Human PSCs	Fan et al., 2023 [39], Chang et al., 2020 [45], Frith et al., 2020 [46], Li et al., 2018 [47], Lai et al., 2017 [48], Schlieve et al., 2017 [49], Workman et al., 2017 [50], Fattahi et al., 2016 [38].
	Mouse gut-derived enteric neural progenitors	Chen et al., 2023 [51], Fujiwara et al., 2022 [52], Nakazawa-Tanaka et al., 2022 [53], Chen 2022 [54], Navoly et al., 2021 [55], Yuan et al., 2021 [56], Bhave et al., 2019 [57], Liu et al., 2018 [58], Findlay et al., 2014 [59], Dettmann et al., 2014 [60], Pan et al., 2011 [61], Mosher et al., 2007 [23], Almond et al., 2007 [25], Natarajan et al., 1999 [22].
	Mouse ESCs	Fujiwara et al., 2022 [62], Hotta et al., 2009 [63], Kawaguchi et al., 2010 [64].
	Mouse other cell sources	Ott et al., 2023 [65], Hu et al., 2019 [66].
	Rat gut-derived enteric neural progenitors	Tian et al., 2021 [67], Zhao et al., 2020 [68], Zhang et al., 2017 [69], Yu et al., 2017 [70]. Tsai et al., 2011 [71].
	Autologous swine gut-derived enteric neural progenitors	Hotta et al., 2023 [33].
	Autologous swine other cell sources	Thomas et al., 2020 [72].
Dysfunctional rat pylorus (gastroparesis)	Rat gut-derived enteric neural progenitors	Dadhich et al., 2020 [73].
	Neural progenitors from non-gut tissues	Micci et al., 2005 [31], Stavely et al., 2022 [74].
nNOS mouse model of colonic dysmotility (chronic constipation)	Mouse gut-derived enteric neural progenitors	McCann et al., 2017 [32], Hotta et al., 2023 [75].

ESCs, embryonic stem cells; nNOS, neuronal nitric oxide synthase; PSCs, pluripotent stem cells.

In addition to the above cell sources, rapidly growing evidence has shown that the enteric glial cells and Schwann cell lineage could also play an important role in replenishing damaged or missing neurons in the intestine [20,76,77]. A number of recent studies have shown that glia have neurogenic potential in response to injury and that transplanted cells give rise to neurons and glia, but the role, function, and neurogenic potential of glial cells post-transplant is yet to be fully elucidated [78,79,80]. Pan et al. [40] recently demonstrated the successful isolation and culture of SCCs residing in the hypertrophic nerve bundles of the aganglionic colon segment of HSCR mice and humans. Using a non-lethal mouse model of colonic aganglionosis generated by diphtheria-toxin-mediated cell specific ablation, the successful survival and engraftment of these HSCR-derived SCCs within the aganglionic gut environment was demonstrated for up to 4 weeks. These transplanted HSCR-SCCs formed neuromuscular connections with recipient aganglionic smooth muscle, leading to improved contractile responses to electrical field stimulation [40].

Clearly, significant progress has been made in preclinical studies with ENSCs, PSCs, and SCCs, with all these cell types demonstrating, to varying degrees, an ability to form ENS. As the field moves from proof-of-concept with in vitro and in vivo animal model studies towards translation to the clinic, it will be interesting to determine which cell type emerges as holding the most ENS therapeutic potential, as well as being safe, readily available, and with protocols for cell generation that are amenable to scale up and good manufacturing practices with compatibility for transplant to patients.

### 2.2. Optimizing Success: Cell Engineering and Modulation of the Gut Environment 

Successful cell therapy for NIDs largely depends on sufficient numbers of cells surviving, engrafting, and differentiating to the appropriate cell types and eliciting the desired functional changes when transplanted into a recipient gut. Therefore, a wide range of approaches have been explored in an effort to maximize the efficacy of cell transplantation, either by engineering the cells themselves, or by modulating the local gut environment into which cells are transplanted. Regarding the former, the overexpression of the anti-apoptotic gene Bcl-2 [81], inhibition of caspase signaling [82], or activation of 5-HT_4_ signaling [83] of donor neural progenitors have all been shown to promote cell survival and differentiation after transplantation into the mouse colon in vivo. Lai et al. used CRISPR/Cas9 gene-editing technology to correct genetic mutations present in HSCR patient-derived iPSC-ENCs, resulting in the normalization of cell migration and differentiation capacity [48]. This suggests that cell engineering by gene editing holds the potential to improve the efficacy of cell therapy for NIDs. In addition, gene editing technology can be used to generate immune-evasive cells by manipulating genes required for immune recognition, such as HLA class I and II proteins [84,85]. This approach could lead to the “Holy grail” of cell therapy—hypoimmunogenic universal donor cell lines. In support of this idea, oligodendrocyte progenitor cells (OPCs) have been developed from healthy human donor-derived iPSCs in which HLA class I and II were knocked out using CRISPR/Cas9. These immune-evading OPCs were transplanted into the brain of a mouse model of Canavan disease, a lethal demyelinating disease caused by mutations in the aspartoacylase (ASPA) gene [86]. The authors demonstrated the low immunogenicity of transplanted OPCs with successful cell survival for up to 6 months along with extensive migration. Transplanted low-immunogenic OPCs were able to differentiate into mature oligodendrocytes and actively remyelinated naked fiber tracks, restored ASPA enzymatic activity, and led to improvement of motor function in recipient mice [86]. Successful transplantation of these universal “off -the-shelf” iPSCs has been shown to have the potential to restore disease phenotypes, highlighting the breakthrough potential of universal donor cells. However, concerns have been raised about the safety of immune cloaking technology in PSCs due to the potential formation of hypoimmunogenic cancers [87]. To reduce such safety risks, universal stem cells could be engineered to contain so-called suicide genes that would allow cells to be eliminated via apoptosis [88]. However, even this approach may not be risk-free as higher mutation rates in malignant cells could potentially cause the loss of suicide genes with the result that even an extremely rare event could lead to the formation of a tumor [88]. Nevertheless, to date, there is limited evidence of progress in immune evasion and safety switch technology in the ENS cell therapy research field.

In addition to modifying donor cells to optimize their ENS restoring ability, several recent papers have deepened our understanding of the effects of transplanted cells on the host’s intestinal environment, opening the door to manipulations here as well. Navoly et al. [55] examined the extracellular matrix (ECM) of recipient gut tissue after ENCDC transplantation and demonstrated the integration of donor cells into the host intestinal tissue and that tissue remodeling was activated by transplanted cells. These observations were translated to a treatment setting in other studies. Yasui et al. [89], using an ex vivo organ culture system, showed that the pre-treatment of recipient gut to degrade ECM proteins enhanced the penetration of transplanted ENCDCs and radial migration within the gut wall. Further, Mueller et al. observed that Agrin, an ECM protein, had an inhibitory effect on ENCDC migration and that cell migration was improved by silencing Agrin expression within enteric neurospheres prior to transplantation to the gut in vivo or in an ex vivo model (Mueller et al., 2023 Stem Cells Transl Med in press). A few early studies have raised the possibilities that the adult gut environment may not be permissive for transplanted embryonic-derived cells to migrate [90]. Furthermore, it has been shown that aged tissue-derived neural progenitors have a significantly reduced capacity to proliferate and migrate compared to fetal cells [21,91]. However, currently there is no available literature directly comparing the ENS forming ability or functional outcomes of cells derived from fetal, post-natal, or adult tissues.

Interestingly, McCann et al. observed that decreased numbers of interstitial cells of Cajal (ICCs) in nNOS-null mice were restored by ENCDC transplantation, potentially contributing to observed improved colonic motility in this model [32]. Similarly, Bhave et al. reported reversed architectural changes in the epithelial layer of an aganglionic colon by the transplantation of ENCDCs [57], suggesting a potential broader application of cell therapy beyond enteric neuropathies or NIDs, such as in inflammatory bowel disease [92] where transplanted cells could also modulate the local gut environment similar to the therapeutic actions of mesenchymal stem cells (MSCs) [93,94,95,96].

### 2.3. Restoration of Gut Function

The ultimate goal of cell replacement therapy for NIDs is to restore intestinal peristalsis and thus ameliorate disease symptoms. Encouragingly, several studies have shown functional improvement in animal models of NIDs following cell transplantation. Transplantation of neural stem cells or ENCDCs into nNOS-null mice, a mouse model of gastroparesis and colonic constipation, has been shown to improve gastric emptying [31] or colonic transit [32,75]. Perhaps due to technical challenges, only recently has it been described that transplanted donor cell-derived neurons can form functional neuromuscular integration. Fattahi et al. co-cultured iPSC-derived enteric neurons and intestinal smooth muscle cells and activated neurons specifically using optogenetic techniques. These authors observed muscle contractions in response to the blue light stimulation of neurons, demonstrating proof-of-concept functional integration between transplanted cell-derived neurons and smooth muscle in an in vitro setting [38]. Subsequently, we have demonstrated neuromuscular integration between transplanted ENCDCs and colonic smooth muscle of a recipient mouse in vivo (Pan et al. unpublished).

One of the most notable achievements in this field over the past few years is the demonstration that the transplantation of iPSC-derived neural crest cells (NCCs) restored peristalsis in the diseased intestine and prolonged the survival of a mouse model of HSCR. Fan et al. [39] generated enteric neural progenitors induced from PSCs. After characterizing these cells using immunohistochemistry, electrophysiology, and transcriptomic profiling, PSC-NCCs were transplanted into Ednrb−/− mice. Interestingly, the authors generated vagal and sacral NCCs separately and reported successful recovery in peristalsis and animal survival only when both types of NCCs were transplanted in vivo [39]. In addition to HSCR, Stavely et al. [74] showed the potential of cell therapy to treat gastroparesis using a novel source of neural progenitors. These authors utilized Schwann cell lineages that reside on peripheral nerve bundles in subcutaneous adipose tissue and that possess characteristics of neural progenitor cells. When these subcutaneous adipose tissue-derived neural stem cells (SAT-NSCs) were transplanted into the stomach of a mouse model of gastroparesis, they showed successful engraftment, migration, and neuroglial differentiation, along with an improvement in gastric emptying [74]. These recent achievements demonstrating functional changes in recipient gut following cell transplant advance the field by providing strong proof-of-concept data for the efficacy of cell therapy to treat NIDs.

### 2.4. Cell Delivery 

Successful cell therapy for NIDs will require targeted delivery of cells to the disease-specific location within the GI tract. Although a few studies have demonstrated that cells injected intraperitoneally can home to the diseased bowel of mice with NIDs [71], in the majority of studies performed to date, cells have generally been surgically delivered to the gut. Either cell suspensions [38,39] or neurospheres [97] have been injected into the gut wall, while other reports have described surgical implantation of neurospheres into the gut wall [29,30,98,99] or their placement onto the serosal surface [32,89,100]. In general, cell injection appears to be a less invasive and more efficient methodology for delivering a larger number of cells than neurosphere implantation surgery. However, side-by-side comparison studies of different delivery methods have not been attempted to date. We recently delivered neurospheres into the colonic wall of nNOS-null mice, a model of delayed gut transit, by single or multiple injections and demonstrated that multiple injections increased cell coverage, leading to improved functional outcomes [75]. These findings suggest that developing and optimizing ways of delivering more cells to the recipient gut, at multiple sites, may be a key factor in maximizing the overall efficacy of cell therapies for NIDs. Interestingly, McCann et al. reported that a single implantation of three neurospheres (~6 × 10^4^ cells in total) covered up to 5 mm^2^ of the gut wall 2 weeks post-transplantation, which was sufficient to reverse the colonic dysmotility of nNOS-null mice [30]. However, it is not clear if a certain area of cell coverage is required to restore gut motility, and establishing dose (number of cells, number of transplant sites, area covered) to function relationships is an important aspect of ongoing research for the field.

In small-animal studies, using rodent models of dysmotility and/or HSCR, the injection or implant approaches outlined above appear to be the most practical, particularly when using early postnatal mice which are small and fragile. However, in translational studies it will be important to establish safe, robust routes and methods of administration that can be validated in a large-animal model prior to clinical use. Endoscopy is one such potential approach as it allows cell delivery to the intestinal wall in a less invasive manner compared to serosal injections which typically require laparotomy. Although we have previously demonstrated the feasibility of endoscopy using HSCR mice [101], we recently performed large-animal (swine) studies using endoscopic ultrasound to target delivery of cell suspensions to the intestinal wall in the region of the myenteric plexus [33]. In these studies, we found that the transplanted cells successfully engrafted within the colonic wall for up to 4 weeks [33]. This first study of its kind, using a clinically relevant endoscopic cell delivery approach in a large-animal model, provides strong support for the technical feasibility and safety of cell delivery to the intestinal wall. 

## 3. Challenges Facing ENS Cell Therapy for the Treatment of Neurointestinal Diseases

### 3.1. Immunological Hurdles: Autologous or Allogeneic

In addition to ensuring that donor cells possess essential cell intrinsic characteristics for ENS restoration, one of the biggest challenges for the clinical application of cell therapy is the immunological rejection of transplanted cells [102]. Some early studies in the central nervous system (CNS), which included human fetal neurotransplantation to Parkinson disease (PD) patients, suggested that the CNS is an immunopriviledged site where allogeneic tissue or cells could be engrafted and remain unaffected by the host’s immunity [103]. Subsequent studies showed that neural stem cells do not express MHC I and II and are therefore less immunogenic and can evade an immunogenic response [104]. Indeed, several clinical trials using allogeneic cells in CNS diseases including PD [105] have been carried out to date, although other clinical trials include immunosuppression [106]. Perhaps unlike the CNS, the GI tract will be an even more challenging organ to target for cell therapy as it harbors a well-developed immune system and 70–80% of immune cells in the body reside in the intestine [107]. Foreign bodies, including exogenously administered (allogeneic) cells not derived from patients themselves may be rapidly eliminated by gut immunity, potentially limiting therapeutic efficacy. Although there are no reports directly comparing the efficacy of autologous and allogeneic ENSC transplants, we have found that autologous transplantation results in better cell engraftment and greater cell coverage than allogeneic transplantation (Ohkura et al. unpublished observations). In terms of clinical development, it may therefore be advantageous to initially pursue autologous cell transplantation in NIDs, particularly in conditions such as HSCR where immunosuppression would be undesirable, until immune evasion or cloaking technologies for iPSC are more fully mature and ready for therapeutic use. 

### 3.2. Establishing a Cell Safety System

To optimize cell-based therapies, the potential of treatment-related toxicities must be considered and minimized. In order to control undesired cell differentiation, toxicity, or ablate engineered cells, a number of “safety switches” have recently been designed and reported as a useful method to manage safety concerns for cell therapy [85]. For example, fluorescein isothiocyanate (FITC) and folate can be conjugated to form a bifunctional safety switch whereby the molecules serve as an “on” switch [108]. In contrast, the herpes simplex virus thymidine kinase (HSV-TK) suicide gene system [109] or inducible caspase 9 (iCasp9)-based safety switch [110] can control the growth and survival of undifferentiated cell populations selectively to reduce the risk of tumor formation while maintaining functional differentiated transplanted cells [111]. Although multiple clinical trials have assessed the safety profile of iCasp9 introduction into T-cell products to treat a variety of diseases (NCT01494103, NCT00710892), and studies involving patients receiving an intravenous injection of iCaps9 cells showed efficient (85–95%) and immediate (within 30 min) elimination of circulating iCasp9-positive cells by administration of a chemical inducer of dimerization [112], no supporting evidence of these types of methodologies is yet available in the ENS cell therapy research field. Further studies will be required to demonstrate that a safety switch strategy can be incorporated into an ENS stem cell therapeutic product for clinical use.

### 3.3. Accessibility Hurdles, Cell Sources

As described above, autologous cells may be superior to allogeneic cells from immunological as well as ENS forming points of view. Moreover, the use of autologous cells does not raise the ethical concerns associated with the derivation of human embryonic stem (ES) cells, thus identifying the optimum tissue source from which to generate autologous cells for clinical application in NIDs is an important consideration. To date, the majority of work on human ENSCs has utilized ENS-containing resected gut tissue as a cell source, but there are some potential drawbacks with harvest of “normoganglionic” tissue using a surgical approach. However, we have shown that autologous neural progenitor cells can be isolated from the aganglionic segment of colon surgically resected during the current standard of care for patients with HSCR [40]. Further, studies have shown that ENCDCs can be isolated from gut mucosal biopsy samples [42], and patients could benefit from this less invasive endoscopic approach for harvesting donor gut tissue. Beyond the gut, derivation of neural stem cells from tissues such as adipose tissue (SAT-NSCs) appears to be feasible [74] and would involve only a minor procedure such as liposuction for tissue harvest. Although autologous cell therapy for NIDs is attractive, and arguably becoming more so due to these more accessible cell sources outlined above, the manufacture of personalized, individual batches of autologous cells at therapeutic grade is currently a costly and time-consuming process. An “off-the-shelf” allogeneic cell product, capable of evading the immune system is the ultimate goal for cell therapy and would have a number of advantages over an autologous product, including immediate availability of cryopreserved batches for patient treatment, standardization of the cell therapy product, and potential lower costs associated with using an industrialized process for production of multiple cell batches [84,85,102].

### 3.4. Where to Start; Target Disease

NIDs encompass a number of diseases that represent a spectrum of enteric neural defects in which the pathophysiology is better characterized in some than in others. In HSCR, for example, the cellular defect (absence of ENS in varying lengths of the distal bowel) is well understood and the extent of aganglionosis is relatively easily determined by analysis of gut mucosal biopsies. In others, such as chronic intestinal pseudo-obstruction, the severity and extent of the neural deficit is less clear, and diagnosis is more reliant on clinical criteria. Perhaps not surprisingly, HSCR is regarded as a good model disease for the generation of proof-of-concept data for cell replacement therapy, not only because of the well-defined cellular deficit in HSCR patients, but also because of the availability of animal models, including rodents with genetic defects that phenocopy the disease (Figure 1). Hence, key preclinical work in a number of labs has focused on the transplant of cells into mice, including the Ednrb−/− mouse model of HSCR, with the aim of rescuing the ENS. Results from these animal studies are encouraging and have shown that transplanted cells engraft, survive, differentiate to enteric neural cell types, extend neurites, form networks, and become functional in the recipient gut in vivo (see Table 1). Ongoing studies are focused on achieving the rescue of gut motile function as assessed by analyzing colonic migrating motor complexes, gut peristalsis, or fecal pellet output as an essential step towards the clinic. Whether this preclinical work can be translated to the development of a cell therapy for HSCR remains to be seen but, notwithstanding some challenges (young pediatric population, variability in extent of aganglionosis patient to patient, other potential changes to the local gut environment), HSCR with its well-defined defects may well be the preferred initial target disease. Alternatives could be NIDs with specific neuronal subtype deficits, such as the loss of nNOS neurons in esophageal achalasia or gastroparesis where the cellular defect may be quite localized. To date, a number of studies using mouse models of gastroparesis have shown that the transplantation of neural progenitors can improve gastric emptying in vivo [31,74]. Achalasia, however, is more challenging from an animal model point of view both in terms of phenocopying the disease and regarding the technical challenges of delivering cells to the lower esophageal sphincter region in a mouse. As many previous studies [113,114,115] have demonstrated, the loss of nitrergic neurotransmission is a major factor in gastroparesis, cell-replacement approaches, either with ENS cells in general, or with a preferential or directed proportion of nNOS neurons, may be a viable therapeutic strategy for this condition. In addition, the impairment of nitrergic innervation in GI sphincter muscles can lead to gastroparesis, esophageal achalasia, and anal achalasia. Therefore, targeting GI sphincters with an inhibitory neuron cell type such as nNOS neurons could also be a viable first-in-human approach, similar to the work conducted with tissue-engineered biosphincters for fecal incontinence (NCT05616208) [116].

### 3.5. Additional Mechanisms of Action (Beyond Cell Replacement)

As we described briefly above, GI abnormalities in HSCR patients may extend beyond the primary defect of hindgut aganglionosis to include the dysfunction of GI smooth muscles and ICC, changes in the extracellular matrix, and in the microbiome, as well as malfunctions of gut immunity. Whether successful replacement of the absent ENS and restoration of gut function by neural cell therapy can also mitigate or repair potential detrimental changes in the local gut environment or in other cell types is still relatively unknown and an area that warrants further investigation. However, although the precise underlying mechanisms are yet to be understood, additional beneficial effects of cell transplantation beyond direct neural cell replacement have been reported in the CNS field [117,118]. NSCs are known to secrete various humoral factors, including neurotrophic factors such as brain-derived neurotrophic factor (BDNF), glial cell-derived neurotrophic factor (GDNF), and nerve growth factor (NGF) [119,120]. It has been suggested that these factors contribute to symptom relief after NSC transplantation to the injured spinal cord [120]. Additionally, IGF-1 and VEGF are also secreted by NSCs and have a role in mitigating the pathophysiology of amyotrophic lateral sclerosis [119]. Furthermore, NSCs have been shown to promote microglial polarization from the proinflammatory M1 phenotype to the anti-inflammatory M2 phenotype and reduce proinflammatory cytokine production [121]. In the nNOS null mouse model of dysmotility, McCann et al. reported a paracrine effect of transplanted ENSCs where a deficit in colonic ICC was restored and which may have helped lead to improved colonic motility [32]. We have also observed decreased colonic inflammation in mice with HSCR after ENSC transplantation (Ohkura et al. unpublished observations), suggesting a possible anti-inflammatory effect of ENSCs. Clearly, there is further scope to better understand, and take advantage of, the potential additional benefits or effects of cell-replacement therapy within the diseased gut environment.

## 4. Conclusions

NIDs continue to represent a range of conditions that are clinically challenging to manage, highlighting the need for new therapies. As the field has made significant progress in demonstrating the potential of cell-based therapies to treat these diseases, opportunities may soon exist to improve the long-term outcomes of patients with NIDs by providing novel, curative cell-replacement approaches.

## Figures and Tables

**Figure 1 biomolecules-14-00229-f001:**
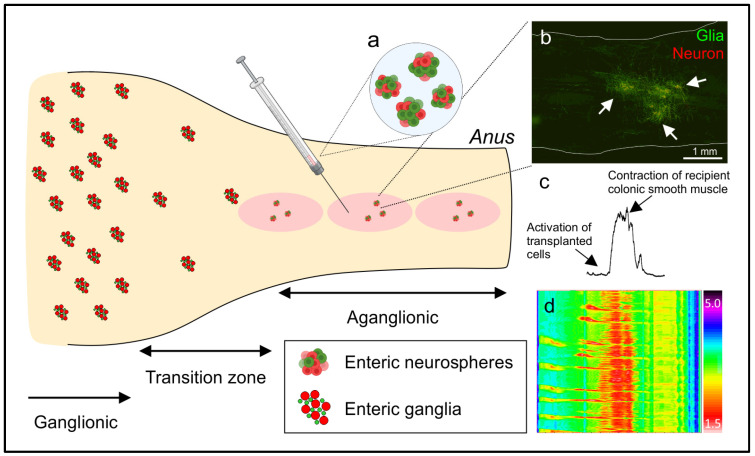
An example of a cell-based treatment strategy for Hirschsprung disease, a congenital neurointestinal disorder. Enteric neurospheres (**a**) containing ENCDCs are injected to site(s) in the diseased (“Aganglionic”) segment of colon where newly generated enteric ganglia ((**b**), arrows) containing functional neurons and glia form. The transplant-derived cells integrate into the neuromuscular circuitry as confirmed with optogenetic activation (**c**), with the result that colonic motility is restored as demonstrated with spatiotemporal maps (**d**).

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
