# Peer review of "Updates and Challenges in ENS Cell Therapy for the Treatment of Neurointestinal Diseases"

_biomolecules, 2024, doi:10.3390/biom14020229_

Round 1

Reviewer 1 Report

Comments and Suggestions for Authors

The review discusses recent progress that has been made in the sourcing and usage of stem cells and the studies into their biology and therapeutic potential for neurointestinal diseases. They also discuss some practical challenges that must be overcome before cell-based therapies can be applied in the clinical setting. Certainly given their track record in the given field they are experts and fully competent. One might criticize that the area irrelatively narrow and specific, however they give a good overview of the area and it is an exciting well written review. I have some minor comments:

- when discussing the hypo-immunogenic cells (line 127) I would at least mention risk associated with immunologically “cloaked” transplants due to the potential formation of hypoimmunogenic cancers.

- in order to support the message I would encourage to have at least one figure and or table included. A figure could visualize the areas that are damaged and where cells where to be seeded. A table could list in an ordered way the experiments done with the information on the animal, disease model and citation e.g. cell type x, used in mouse model of Hirschsprung disease, positive outcomes + limitations, citation.

- line 134 Transplanted low-immunogenic OPCs were remyelinated, restored ASPA enzymatic activity and led to improvement of motor function in recipient mice. Must be corrected to "...were able to differentiate in mature oligodendrocytes and actively remyelinating naked fiber tracks..."

Author Response

The review discusses recent progress that has been made in the sourcing and usage of stem cells and the studies into their biology and therapeutic potential for neurointestinal diseases. They also discuss some practical challenges that must be overcome before cell-based therapies can be applied in the clinical setting. Certainly given their track record in the given field they are experts and fully competent. One might criticize that the area irrelatively narrow and specific, however they give a good overview of the area and it is an exciting well written review. I have some minor comments:

1. when discussing the hypo-immunogenic cells (line 127) I would at least mention risk associated with immunologically “cloaked” transplants due to the potential formation of hypoimmunogenic cancers.

We appreciate this comment from the reviewer. Sentences describing the risk associated with the use of hypoimmunogenic cells have been added to Section 2.2 Optimizing success: cell engineering and modulation of the gut environment (Lines 144-150).

2. In order to support the message I would encourage to have at least one figure and or table included. A figure could visualize the areas that are damaged and where cells where to be seeded. A table could list in an ordered way the experiments done with the information on the animal, disease model and citation e.g. cell type x, used in mouse model of Hirschsprung disease, positive outcomes + limitations, citation.

Thank you for such an important suggestion. We generated a schematic diagram, showing an example of a strategy for using neural stem cells to treat the enteric neuropathy Hirschsprung disease, which has been added to the Section 3.4. Where to start: target disease in the revised manuscript.

We also compiled a table, summarizing a number of published studies in the ENS cell therapy research field using different cell sources and various animal models of neurointestinal diseases. The new table has been added to the Section 2.1. Cell sources.

3. line 134 Transplanted low-immunogenic OPCs were remyelinated, restored ASPA enzymatic activity and led to improvement of motor function in recipient mice. Must be corrected to "...were able to differentiate in mature oligodendrocytes and actively remyelinating naked fiber tracks..."

Thank you. The sentence has been revised accordingly (Lines 140-141 in the revised manuscript).

Reviewer 2 Report

Comments and Suggestions for Authors

In their review article, Ohkura et al discusses the therapeutic potential of stem cells in addressing neurointestinal diseases. The authors have presented a systematic review with focus on sources of putative stem cells, factors regulating optimum gut environment for maximum efficacy and transplantation, mechanisms of cell delivery and associated immunological hurdles.

Major comments/ suggestions

1.       It would be informative to highlight the enteric neuron- enteric glia composition in the ENS and discuss the role of enteric glial cells in enteric neurogenesis during injury, regeneration and disease.

2.       It is pertinent to mention the neuro- and muscular aspects that regulate gut motility functions, and how neuronal stem cells influence smooth muscle functions.

3.       The authors could include a section on the disorders & success in pediatric versus adult stage in animal models/ humans, and the success of transfer in embryonic versus postnatal scenarios in animal models.  It is also recommended to add a short description on the ENS stem cell markers.

4.       The authors could also add a note on the genetic regulation of enteric neuropathies, and discuss briefly on how advances in single cell transcriptomics of mouse and human ENS could help understand the enteric neuropathies better, and help elucidate the right target diseases for stem cell therapy.

Author Response

In their review article, Ohkura et al discusses the therapeutic potential of stem cells in addressing neurointestinal diseases. The authors have presented a systematic review with focus on sources of putative stem cells, factors regulating optimum gut environment for maximum efficacy and transplantation, mechanisms of cell delivery and associated immunological hurdles.

Major comments/ suggestions

  1. It would be informative to highlight the enteric neuron- enteric glia composition in the ENS and discuss the role of enteric glial cells in enteric neurogenesis during injury, regeneration and disease.

Although this is a very important topic, we believe a summary of this field is beyond the scope of our current article and could even merit a dedicated review. Therefore, we decided to add a brief summary of the previous work showing neurogenic potential of enteric glial cells. New sentences with a few new references have been added to Section 2.1. Cell sources (Lines 99-102).

  1. It is pertinent to mention the neuro- and muscular aspects that regulate gut motility functions, and how neuronal stem cells influence smooth muscle functions.

We appreciate this comment from the reviewer. Sentences mentioning neuromuscular connectivity have been added to the paragraphs in the section 2.3. Restoration of gut function (Lines 186-194) of the revised manuscript.

  1. The authors could include a section on the disorders & success in pediatric versus adult stage in animal models/ humans, and the success of transfer in embryonic versus postnatal scenarios in animal models.  It is also recommended to add a short description on the ENS stem cell markers.

We appreciate these important suggestions. There is some data comparing the behaviour of cells that were transplanted into fetal, postnatal or adult recipients. There is also some literature demonstrating embryonic or postnatal cell transplants to animal models. However, in these studies we noted the lack of comparisons of ENS forming success or functional outcomes in the above combinations. Therefore, we have added a brief summary of the literature, which has been added to Section 2.2. Optimizing success: cell engineering and modulation of the gut environment (Lines 165-171).

  1. The authors could also add a note on the genetic regulation of enteric neuropathies, and discuss briefly on how advances in single cell transcriptomics of mouse and human ENS could help understand the enteric neuropathies better, and help elucidate the right target diseases for stem cell therapy.

Our understanding of Hirschsprung disease as a complex genetic disorder has been significantly advanced over the last 30 years whereas the cell replacement therapeutic approach we and others propose is not dictated by the genetic status of the patient. Single cell transcriptomics of mouse and human ENS has recently been introduced to the research community and rapidly expanded our understanding of cell biology and development. In the current special issue, new insights of the pathogenesis of neurointestinal diseases have been discussed by other authors and therefore this topic is not within the scope of this particular review article.

Reviewer 3 Report

Comments and Suggestions for Authors

The current review is well written, concise, and engaging, focusing on a topic of utmost significance in the field of medicine, particularly in gastroenterology. The topic, as suggested by the title, revolves around the updates and challenges in enteric nervous system (ENS) cell therapy for the treatment of neurointestinal diseases.

In the introduction, the prevalence and social costs of the so-called neurointestinal diseases together with general data on cell therapy are clearly presented. In chapter two, the authors introduce the different cell sources, which could be used for ENS based cell therapy. Classical enteric neuronal stem cells but also nowadays very used IPSCs and Schwann cells are discussed as possible sources. Precisely, the authors provide examples of successful preclinical animal experiments using these different cell sources. The manuscript than, describe the importance of cell engineering, on one side, and modulation of the intestinal environment, on the other side, to optimize the cell therapy technique. The authors articulate that the final objective of this technique is not solely to recover the absent or defected ENS cells, but also to reinstate its functions, such as gastric emptying and intestinal peristalsis. They mention the different methods of cell delivery, discussing the importance of providing data from large animal models.  Here they cite their recently high-level published work using the pig as a model for endoscopic delivery of autologous enteric neuronal stem cells. The third chapter focuses on the challenges facing ENS cell therapy and on the additional factors, which can increase or decrease its success rate. 

I believe this review is crucial for delineating the status of experiments in this field, identifying areas that require further research and highlighting the obstacles that need to be considered doing it.

I only have a few minor comments:

·         Maybe the authors could try to find newer information about the annual cost of functional digestive diseases mentioned in the introduction. The actual reference is dated 1998 there are surely more recent estimations.

·         Just as suggestion: Data about the number of cells needed to restore the functions could be interesting. For instance, is the number of implanted cells important in order to restore gut functions?

·         Another suggestion: The autologous cell transplantation has an advantage also from the ethical point of view, this could be added.

·         Is in the literature any example of ENS cell therapy with allogenic transplantation?

·         The abbreviation NID is not introduced

·         Line 22: The abbreviation GI is introduced twice.

·         Line 69: here you can use the abbreviation for Hirschsprung disease that should be introduced at line 26

·         Line 152: Provide the definition for nNOS abbreviation.

·         Line 109: Provide the definition for GMP abbreviation.

·         Line 244: Explain abbreviation FITC.

·         Line 328: Provide the definition for all abbreviations.

Author Response

The current review is well written, concise, and engaging, focusing on a topic of utmost significance in the field of medicine, particularly in gastroenterology. The topic, as suggested by the title, revolves around the updates and challenges in enteric nervous system (ENS) cell therapy for the treatment of neurointestinal diseases.

In the introduction, the prevalence and social costs of the so-called neurointestinal diseases together with general data on cell therapy are clearly presented. In chapter two, the authors introduce the different cell sources, which could be used for ENS based cell therapy. Classical enteric neuronal stem cells but also nowadays very used IPSCs and Schwann cells are discussed as possible sources. Precisely, the authors provide examples of successful preclinical animal experiments using these different cell sources. The manuscript than, describe the importance of cell engineering, on one side, and modulation of the intestinal environment, on the other side, to optimize the cell therapy technique. The authors articulate that the final objective of this technique is not solely to recover the absent or defected ENS cells, but also to reinstate its functions, such as gastric emptying and intestinal peristalsis. They mention the different methods of cell delivery, discussing the importance of providing data from large animal models.  Here they cite their recently high-level published work using the pig as a model for endoscopic delivery of autologous enteric neuronal stem cells. The third chapter focuses on the challenges facing ENS cell therapy and on the additional factors, which can increase or decrease its success rate. 

I believe this review is crucial for delineating the status of experiments in this field, identifying areas that require further research and highlighting the obstacles that need to be considered doing it.

I only have a few minor comments:

  1. Maybe the authors could try to find newer information about the annual cost of functional digestive diseases mentioned in the introduction. The actual reference is dated 1998 there are surely more recent estimations.

Thank you. We have cited a few more recent articles to discuss the impact of NIDs on the world population as well as health care has been added to Introduction section (Lines 29-33) in the revised manuscript.

  1. Just as suggestion: Data about the number of cells needed to restore the functions could be interesting. For instance, is the number of implanted cells important in order to restore gut functions?

We appreciate this important suggestion. Sentences discussing possible number of cells required to restore gut motility function has been added to the Section 2.4. Cell delivery (Lines 228-233) in the revised manuscript.

  1. Another suggestion: The autologous cell transplantation has an advantage also from the ethical point of view, this could be added.

We agree with the comment from this reviewer. A sentence of an advantage of autologous cells from ethical point of view has been added to the Section 3.3. Accessibility hurdles, cell sources (Lines 293-294) in the revised manuscript.

  1. Is in the literature any example of ENS cell therapy with allogenic transplantation?

Most of the studies conducted to date have been with mouse cells transplanted to mouse littermates, or to mice of the same or different strains. Human cells have also been transplanted to mouse recipients so in fact the majority of transplants have been allogenic. There are only a few examples of true autologous transplants which were performed in swine and are cited in Table 1.

  1. The abbreviation NID is not introduced.

The abbreviation, NIDs is introduced in the line 26.

  1. Line 22: The abbreviation GI is introduced twice.

Thank you. The sentence has been revised accordingly.

  1. Line 69: here you can use the abbreviation for Hirschsprung disease that should be introduced at line 26

The abbreviation, HSCR is now introduced in the line 26, instead of Line 69.

  1. Line 152: Provide the definition for nNOS abbreviation.

The abbreviation, nNOS has been added to the footnote of the Table 1.

  1. Line 109: Provide the definition for GMP abbreviation.

Thank you. Abbreviation, GMP has been removed and spelled out in the revised manuscript.

  1. Line 244: Explain abbreviation FITC.

The abbreviation, FITC has been added in the revised manuscript.

  1. Line 328: Provide the definition for all abbreviations.

Thank you, all abbreviations have been provided in the revised manuscript.
